# Combining DNA Barcoding and HPLC Fingerprints to Trace Species of an Important Traditional Chinese Medicine Fritillariae Bulbus

**DOI:** 10.3390/molecules24183269

**Published:** 2019-09-08

**Authors:** Yingchun Zhong, Haiying Wang, Qianhe Wei, Rui Cao, Hailong Zhang, Yongzhi He, Lizhi Wang

**Affiliations:** School of Chinese Materia Medica, Tianjin University of Traditional Chinese Medicine, Tianjin 301617, China

**Keywords:** DNA barcoding, Fritillariae Bulbus, ITS, ITS2, HPLC

## Abstract

Fritillariae Bulbus is a precious Chinese herbal medicine that is grown at high elevation and used to relieve coughs, remove phlegm, and nourish the lungs. Historically, Fritillariae Bulbus has been divided into two odourless crude drugs: Fritillariae Cirrhosae Bulbus and Fritillariae Thunbergii Bulbus. However, now the Chinese Pharmacopoeia has described five Fritillariae Bulbus—the new additions include Fritillariae Pallidiflorae Bulbus, Fritillariae Ussuriensis Bulbus, and Fritillariae Hupehensis Bulbus. Because the morphology of dried Fritillariae Bulbus is similar, it is difficult to accurately identify the different types of Fritillariae Bulbus. In the current study, we develop a method combining DNA barcoding and high-performance liquid chromatography (HPLC) to help distinguish Fritillariae Cirrhosae Bulbus from other Fritillariae Bulbus and guarantee species traceability of the five types of Fritillariae Bulbus. We report on the validation of an integrated analysis method for plant species identification using DNA barcoding that is based on genetic distance, identification efficiency, inter- and intra-specific variation, calculated nearest distance, neighbour-joining tree and barcoding gap. Our results show that the DNA barcoding data successfully identified the five Fritillariae Bulbus by internal transcribed spacer region (ITS) and ITS2, with the ability to distinguish the species origin of these Fritillariae Bulbus. ITS2 can serve as a potentially useful DNA barcode for the *Fritillaria* species. Additionally, the effective chemical constituents are identified by HPLC combined with a chemical identification method to classify *Fritillaria*. The HPLC fingerprint data and HCA (hierarchical clustering analysis) show that Fritillariae Cirrhosae Bulbus is clearly different from Fritillariae Thunbergii Bulbus and Fritillariae Hupehensis Bulbus, but there is no difference between Fritillariae Cirrhosae Bulbus, Fritillariae Ussuriensis Bulbus, and Fritillariae Pallidiflorae Bulbus. These results show that DNA barcoding and HPLC fingerprinting can discriminate between the five Fritillariae Bulbus types and trace species to identify related species that are genetically similar.

## 1. Introduction

Fritillariae Bulbus (Bei-mu in China) is a well-known traditional Chinese medicine that has been known for more than 2000 years for its ability to moisten the lungs, resolve phlegm, and relieve coughs [1]. Historically, Fritillariae Bulbus is divided into two odourless crude drugs, namely, Fritillariae Cirrhosae Bulbus and Fritillariae Thunbergii Bulbus. The Chinese Pharmacopoeia has described five Fritillariae Bulbus, including Fritillariae Cirrhosae Bulbus, Fritillariae Ussuriensis Bulbus, Fritillariae Pallidiflorae Bulbus, Fritillariae Thunbergii Bulbus, and Fritillariae Hupehensis Bulbus, which come from 11 unique plants. According to the Chinese Pharmacopoeia, Fritillariae Cirrhosae Bulbus originates from six plants, including *Fritillaria cirrhosae* D. Don, *Fritillaria unibracteata* Hsiao et K. C. Hsia, *Fritillaria przezvalskii* Maxim, *Fritillaria delavayi* Franch, *Fritillaria taipaiensis* P. Y. Li, and *Fritillaria unibracteata* Hsiao et K. C. Hsia var. *wabuensis* Z. D. Liu, S. Wang et S. C. Chen [2]. The primary plants of Fritillariae Pallidiflorae Bulbus are *Fritillaria walujewii* and *Fritillaria pallidiflora* Schrenk, which are mainly distributed in the Xinjiang province [3]. Fritillariae Ussuriensis Bulbus, Fritillariae Thunbergii Bulbus, and Fritillariae Hupehensis Bulbus are from *Fritillaria ussuriensis* Maxim, *Fritillaria thunbergii* Miq, and *Fritillaria hupehensis* Hsiao et K. C. Hsia, respectively. Wild Fritillariae Cirrhosae Bulbus, which has excellent quality and good curative effects, is distributed between the altitudes of 3000 and 5000 m in most parts of Tibet, Northwest Sichuan, Northern Yunnan, Southeastern Qinghai, and Eastern Europe [4]. Because the price of Fritillariae Cirrhosae Bulbus has reached 3000 yuan per kilogram [5], which is higher than other Fritillariae Bulbus, merchants adulterate other Fritillariae Bulbus to sell as Fritillariae Cirrhosae Bulbus to obtain high profits. Unfortunately, differentiating between Fritillariae Bulbus types is difficult because of their similar morphological characteristics. Thus, to control the quality of herbs and ensure the drug safety of Fritillariae Bulbus, an accurate and efficient identification method should be urgently explored. Previous studies have suggested that Fritillariae Bulbus are rich in a variety of alkaloids, which are effective ingredients for treating diseases and have been confirmed by animal testing [6,7,8]. The results showed that steroidal alkaloids from Fritillariae Bulbus, such as imperialine, peimisine, verticinone, and verticine, relaxed the bronchial smooth muscle [9], significantly inhibited the proliferation and colony formation of glioblastoma multiforme cells, and aided in the treatment of mastitis by reducing histopathological lesions of the breast [10,11].

In recent years, identification methods for Fritillariae Cirrhosae Bulbus have been developed, including the determination and identification of total alkaloids by near-infrared spectroscopy. Microscopic identification was carried out by observing the characteristics of starch granules and the shape of the umbilical point of starch granules in bulbs. High-performance liquid chromatography (HPLC) was used to identify isosteroidal alkaloids [12,13,14]. However, 10 crude herbs used in the study contained imperialine, peimisine, verticinone, and verticine, making the HPLC method insufficient. These previous studies successfully determined the contents of imperialine, peimisine, verticinone, and verticine for the quality evaluation of Fritillariae Bulbus using HPLC. However, these methods are still inadequate for distinguishing the species of Fritillariae Bulbus.

There is a foundation for distinguishing the authenticity of various plants and animals by establishing DNA barcodes. Scientist Chen et al. uses DNA sequences as a taxonomic basis to design barcodes [15,16,17,18,19]. The regularity of cytochrome C oxidase I (COI) sequence differentiation enables many kinds of animals and their related species to be recognized [20,21,22,23]. Plant mitochondrial genome genetic differentiation is small, so COI is not suitable for plants. Yao et al. evaluated the ITS2 barcodes from 50,790 plants and 12,221 animals [24]. Scientists have used rDNA-internal transcribed spacers (ITS), *rbcL*, *matK*, *psbA*-*trnH,* and *trnL-F* to identify plant species and analyse phylogenetic relationships [25,26,27,28]. Because of the difference in amplification efficiency among species and the different needs of primer generality, barcodes were screened by combining genes. The chloroplast genome was sequenced, and phylogenetic analysis was used to evaluate the phylogenetic relationships among species and determine whether phylogenetic relationships were consistent with their geographical distribution patterns. Moon et al. researched the markers in *matK* and *RPS16* to distinguish five Fritillariae species [29]. Wang et al. used polymerase chain reaction restriction fragment length polymorphism (PCR-RFLP) to identify Fritillariae Cirrhosae Bulbus [30]. They could distinguish between species, but could not distinguish within species. The clustering results showed that the geographical distribution of *Fritillaria* resources was similar to that of Fritillariae Cirrhosae Bulbus [31,32,33,34].

To control the quality of crude drugs, it is necessary to confirm their identities and ensure the species traceability of Fritillariae Bulbus. In this study, we tested four candidate loci, namely, the chloroplast gene regions *matK*, *psbA*-*trnH*, *ITS,* and ITS2, for their validity as DNA barcodes to identify species in Chinese medicinal Fritillariae Bulbus. To establish a method that enables the species traceability of Fritillariae Bulbus, we performed experiments to quantify four chemical markers using HPLC/ELSD.

## 2. Results

### 2.1. Efficiency of PCR Amplification and Sequencing

The three loci—ITS, ITS2, and *psbA*-*trnH*—all showed PCR amplification and high sequencing efficiency (100%). The locus *matK* showed no PCR amplification and a sequencing efficiency of 0%. Moreover, the effective sequence ratios of ITS, ITS2, and *psbA*-*trnH* were the same (100%).

### 2.2. Genetic Divergence Determination

The lengths of the aligned sequence (base pairs) and variable sites for the ITS and ITS2 regions were 610/46 and 235/18, respectively. The results are shown in Figure 1 and Figure 2. Six parameters were used to characterise inter- and intraspecific divergence. The intraspecific distance and interspecific distance of ITS were 0.0000–0.0145 and 0.0020–0.0052, respectively, and the average distance was 0.0024. For ITS2, the intraspecific distance was 0.0000–0.0085, the interspecific distance was 0.0021–0.0053, and the average distance was 0.0030. The results indicated that ITS2 had the highest interspecific divergence, followed by ITS. Meanwhile, ITS exhibited lower interspecific divergence compared with the other regions. At the intraspecific level, ITS showed the lowest divergence, while ITS2 displayed the highest variation level. Basic local alignment search tool (BLAST) analyses were performed on the 88 sequences. The results indicated that compared with the same sequence in NCBI, query cover was over 99%.

### 2.3. Neighbour-Joining (NJ) Tree Identification

The NJ tree constructed from 44 ITS sequences showed that Fritillariae Cirrhosae Bulbus could be clustered into one group, Fritillariae Thunbergii Bulbus and Fritillariae Hupehensis Bulbus could be clustered into one group, Fritillariae Pallidiflorae Bulbus could be clustered into one group, and Fritillariae Ussuriensis Bulbus could be clustered into one group. The results are shown in Figure 3.

### 2.4. Cluster Analysis by High-Performance Liquid Chromatography

To validate the results of the HPLC fingerprint analysis and further elucidate the resemblance relationship among the samples, hierarchical clustering analysis (HCA) was applied using the Statistical Product and Service Solutions (SPSS)19.0 software and the Unscrambler X 10.0 software. HCA results showed that *Fritillaria cirrhosae* was clustered into a group of circle samples with *Fritillaria ussuriensis* as black marker. The results are shown in Figure 4. Fritillariae Thunbergii Bulbus and Fritillariae Hupehensis Bulbus were clearly distributed on both the sides by HPLC. Fritillariae Pallidiflorae Bulbus were dispersed in the same side of Fritillariae Cirrhosae Bulbus, and HPLC could not distinguish Fritillariae Cirrhosae Bulbus from other *Fritillaria.* The alkaloid content is shown in Appendix A. The chemical constituents of *Fritillaria* were similar. The results were consistent with the results of fingerprint analysis by HPLC. Therefore, HCA can also help to distinguish *Fritillaria* plants, but it is not enough for identification.

A total of 44 Fritillariae Bulbus medicines were analysed by chromatographic fingerprints and a principal component diagram. All samples were divided into three parts. SPSS 19.0 was used for systematic clustering, and the clustering method of inter-group connection was adopted. The measurement standard was the squared Euclidean distance, and the effective percentage of samples was 100 percent. Chromatographic data showed that Fritillariae Thunbergii Bulbus, Fritillariae Hupehensis Bulbus, Fritillariae Ussuriensis Bulbus, and Fritillariae Pallidiflorae Bulbus had similar fingerprints. The Fritillariae Cirrhosae Bulbus samples were clustered into one branch, and the counterfeit products were clearly on different branches. Fritillariae Cirrhosae Bulbus and Fritillariae Thunbergii Bulbus had similar fingerprints. These results showed that HPLC cannot distinguish the two kinds of traditional Chinese medicines. The results are shown in Figure 5.

## 3. Discussion

### 3.1. Species Resources and DNA Barcodes of Fritillariae Bulbus

There are many studies on the Fritillariae Cirrhosae Bulbus plant. However, for the high circulation of traditional Chinese medicine on the market, DNA barcodes identification of Fritillariae Bulbus should be studied. Fritillariae Cirrhosae Bulbus in the 2015 edition of Chinese Pharmacopoeia can be used for medicinal purposes, and *Fritillaria* species have similar morphologies. Wild Fritillariae Bulbus grows in high altitude areas and is difficult to harvest, requiring a large amount of manpower and material resources. It is very important to identify Fritillariae Bulbus effectively. It is difficult to distinguish Fritillariae Cirrhosae Bulbus from other Fritillariae Bulbus medicinal materials with traditional methods. Powder identification is also a common method for the identification of traditional Chinese medicines. Fritillariae Cirrhosae Bulbus and other Fritillariae Bulbus medicinal materials are white, tasteless powders. The classification of Fritillariae Cirrhosae Bulbus medicinal materials has always been controversial, and there are differences among different types of Fritillariae Cirrhosae Bulbus medicinal materials in the Chinese Pharmacopoeia. DNA barcode identification may be useful in the classification of *Fritillaria* parent sources. The ITS and ITS2 gene regions have been written into the Chinese Pharmacopoeia and have been widely used to identify plants of the Fabaceae and Compositae families as well as other plants that are traditionally used in Chinese medicines. ITS and ITS2 have good amplification efficiency for the identification of Fritillariae Cirrhosae Bulbus. *PsbA*-*trnH* also exhibits sufficient amplification efficiency in Fritillariae Cirrhosae Bulbus, but it is difficult to distinguish Fritillariae Cirrhosae Bulbus and its counterfeits accurately. The locus *matK* cannot be efficiently amplified in Fritillariae Cirrhosae Bulbus. 

### 3.2. Identification of Fritillariae Bulbus

ITS and ITS2 can clearly distinguish the five Fritillariae Bulbus types. In the NJ tree, Fritillariae Bulbus can be clearly divided into different branches. The same results can be obtained by using HPLC to identify Fritillariae Bulbus. Fritillariae Cirrhosae Bulbus (six dry bulbs from Liliaceae, *Fritillaria cirrhosae* D. Don, *Fritillaria unibracteata* Hsiao et K. C. Hsia, *Fritillaria przezvalskii* Maxim, *Fritillaria delavayi* Franch, *Fritillaria taipaiensis* P. Y. Li, and *Fritillaria unibracteata* Hsiao et K. C. Hsia var. *wabuensis* Z. D. Liu, S. Wang et S. C. Chen) can be divided into one branch, and the rest of the Fritillariae Bulbus can be divided into different branches. DNA barcodes can not only identify the original medicinal materials but also identify the different parts and residues of medicinal materials. Additionally, we utilized HPLC fingerprint and HCA methods to identify the five Fritillariae Bulbus. The data showed clear differences in the HPLC fingerprints and HCA, allowing for the differentiation of Fritillariae Cirrhosae Bulbus, Fritillariae Thunbergii Bulbus, and Fritillariae Hupehensis Bulbus. However, Fritillariae Cirrhosae Bulbus, Fritillariae Ussuriensis Bulbus, and Fritillariae Pallidiflorae Bulbus could not be distinguished using HPLC fingerprint and HCA methods.

In summary, we have established new chemical and molecular analysis methods for discriminating the five Fritillariae Bulbus. The results revealed that DNA barcoding overcomes the limitations of HPLC fingerprinting for differentiating close genetic relationships and similar chemical-composition species to guarantee an accurate and scientific confirmation of herbal identities in medicinal materials from multiple sources.

## 4. Materials and Methods

### 4.1. Sampling

In total, 44 samples representing 10 *Fritillaria* species were collected from different areas of China. All of the plant species were identified by Lijuan Zhang. Detailed information about the materials used and sequences obtained in the study are provided in Appendix A.

### 4.2. DNA Extraction, Amplification, and Sequencing

Genomic DNA was extracted from dried Bulbus samples using a Plant Genomic DNA Extraction Kit (Tiangen Biotech, Beijing, China) according to the manufacturer’s protocol. The quality of the extracted DNA samples was verified to ensure their suitability for subsequent amplification. The four candidate barcodes regions (ITS, *matK*, *psbA*-*trnH*, and ITS2) were tested for their feasibility for *Fritillaria* barcoding.

The PCR products were detected by 1.2% agarose gel electrophoresis and visualized under ultraviolet light. DNA extraction, amplification, and sequencing were carried out with identical procedures and conditions described above. Forty-four sequences from five Fritillariae Bulbus were submitted in Genbank (Appendix A).

### 4.3. Sequence Analysis

DNA sequences were aligned with NCBI standard sequences by using the software Clustal W [35] and MegaⅩ and assembled with CodonCode Aligner version 3.7.1 (CodonCode, USA). Removal of the primer region and low-quality region, manual correction, and stitching were performed. In addition, all of the ITS sequences in our study were delimited based on hidden Markov model (HMM) annotation methods. The average intra- and inter-specific distances were calculated to evaluate the intraspecies variation and interspecific divergence [36,37]. Cluster analyses were conducted using a neighbour-joining tree based on the pairwise Kimura 2-parameter model between individuals.

### 4.4. Chromatographic Conditions

#### 4.4.1. Column and Instrument Conditions

The instrument was fitted with a Waters ACQUITY UPLC^TM^ BEH C18 column (100 mm × 2.1 mm, 1.7 µm); mobile phase, acetonitrile (A), and 0.02% aqueous triethylamine (B); flow rate, 0.25 mL/min; column temperature, 25 °C; injection volume, 1 µL; and gradient elution program (0–3 min, 30–60% A; 3–4 min, 60–80% A; and 4–10 min, 80–100% A). The parameters of the evaporative light scattering detector were set as follows: Drift tube temperature, 40 °C; sprayer parameters, 40%; gain value, 500; and gas pressure, 30 psi.

#### 4.4.2. Preparation of the Reference Solution

Two milligrams of each alkaloid reference substance, peimisine (1), verticine (2), verticinone (3), and imperialine (4), were precisely weighed and placed in a 2 mL flask. A known amount of methanol was added to the vial, and the mixture was shaken to ensure uniformity of the prepared mixed reference solution.

#### 4.4.3. Preparation of Test Solution

Approximately two grams of medicinal powder (sieve 4) was precisely weighed, placed in a 100 mL flask, and soaked in a 4 mL aqueous ammonia solution for 2 hours. The solution was then mixed with 60 mL of a mixed solution of trichloromethane-methanol (4/1) and homogenized and refluxed for 3 hours in a water bath at 80 °C before cooling and filtering. The filtrate was placed in an evaporating dish and allowed to dry. The sample was reconstituted in methanol and transferred to a 2 mL volumetric flask. Methanol was added to a known amount, and the mixture was shaken. The solution was filtered through a 0.22 µm microporous membrane filter before HPLC analysis.

#### 4.4.4. Methodological Investigation

To determine the precision, three solutions of reference substances with different concentrations were sampled in triplicate. The retention time (RT) and peak area (PA) of each peak were measured, and RSD values were calculated. The retention times and peak areas of fritillarin, sibelin, fritillarin B, and fritillarin A had RSD values ranging from 0.12% to 1.09% and from 0.68% to 3.84%, respectively.

Reproducibility was established by extracting six samples of Fritillariae Thunbergii Bulbus and Fritillariae Cirrhosae Bulbus from the same batch according to the abovementioned methods. The peak area of each reference substance was determined in the range of 0.85–2.75% according to the chromatographic conditions described above, which showed that the method had good reproducibility.

In the stability experiment, the sample solutions were analysed at 0, 2, 4, 8, 12, 24, and 36 hours. The peak area was recorded. The RSD values of the corresponding concentration of the peak area of each reference substance ranged from 0.71% to 3.47%, indicating that the sample was stable for at least 36 hours.

Sample determination: a total of 44 batches of 10 *Fritillaria* medicinal materials were sampled and analysed according to the preparation method of the solution. Sample analysis was carried out according to the chromatographic conditions described above. The sample injection volume was 1 µL, and chromatographic charts were recorded for 10 min. Meanwhile, the reference solution was injected to determine the retention time of alkaloids. Representative HPLC chromatograms of 10 *Fritillaria* medicinal materials are shown in Appendix A.

## Figures and Tables

**Figure 1 molecules-24-03269-f001:**
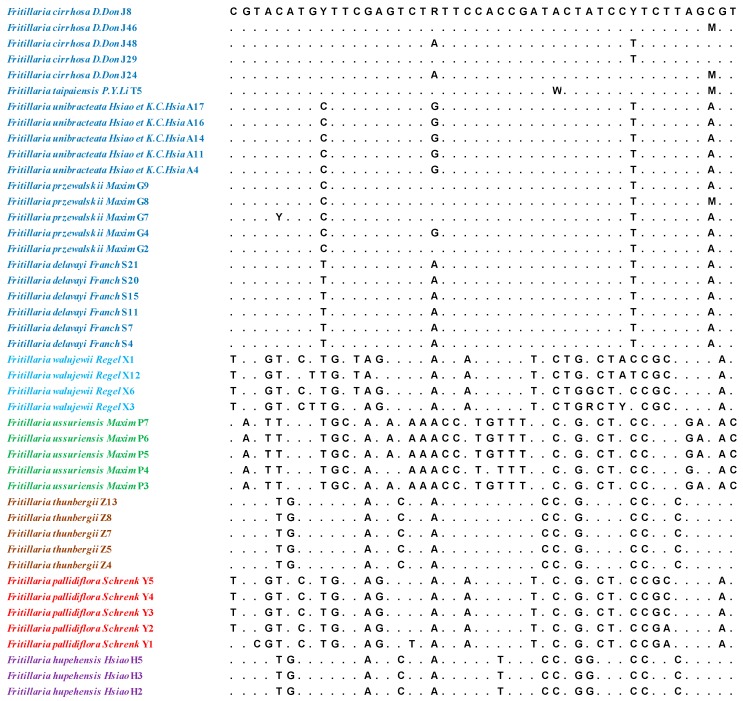
The intraspecific variable sites in the internal transcribed spacer region (ITS) sequences of Fritillariae Bulbus.

**Figure 2 molecules-24-03269-f002:**
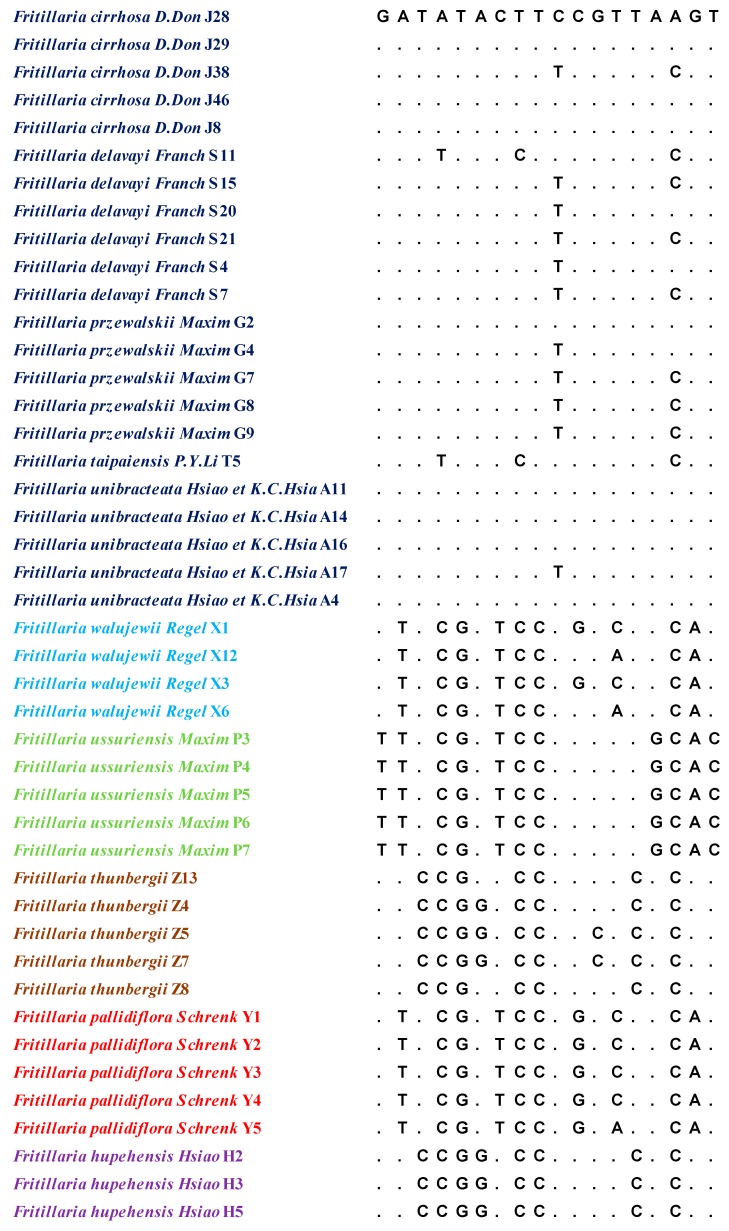
The intraspecific variable sites in the ITS2 sequences of Fritillariae Bulbus.

**Figure 3 molecules-24-03269-f003:**
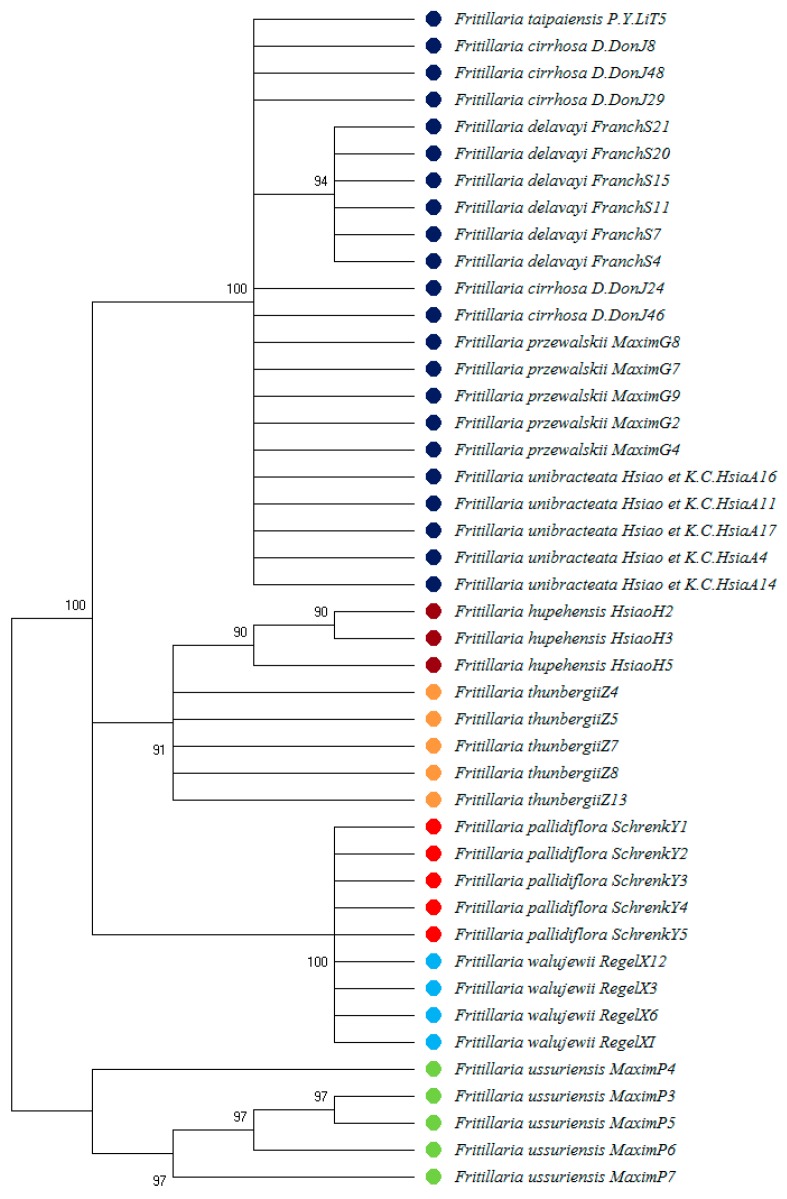
Neighbour-joining (NJ) tree based on ITS sequences.

**Figure 4 molecules-24-03269-f004:**
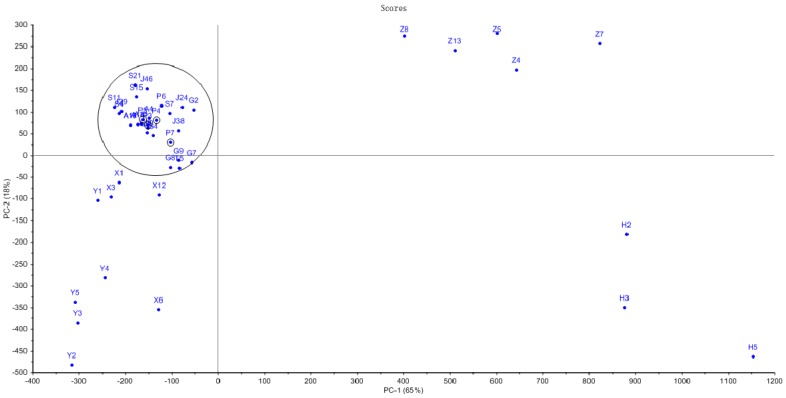
The Unscrambler of high-performance liquid chromatography (HPLC) chemometrics analysis of 44 samples of Fritillariae Bulbus.

**Figure 5 molecules-24-03269-f005:**
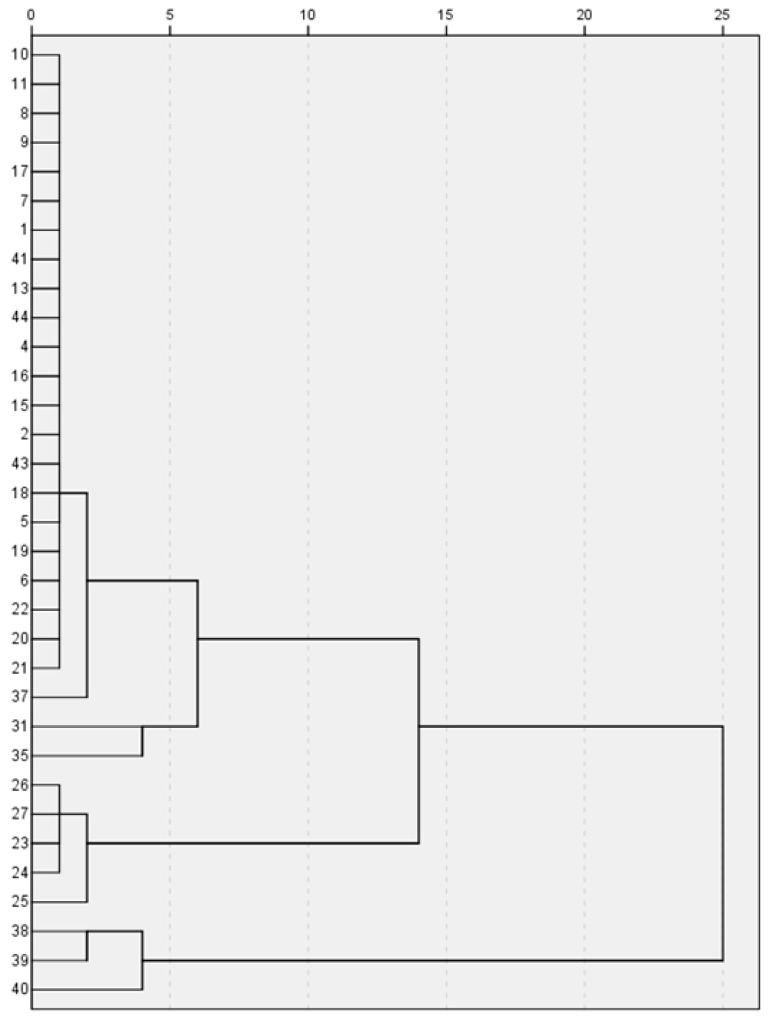
Dendrogram for hierarchical clustering by Statistical Product and Service Solutions (SPSS) of 44 samples of Fritillariae Bulbus.

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
