# Peer review of "Combining DNA Barcoding and HPLC Fingerprints to Trace Species of an Important Traditional Chinese Medicine Fritillariae Bulbus"

_molecules, 2019, doi:10.3390/molecules24183269_

Round 1
Reviewer 1 Report
Can the authors explain why the following items were not found in the cited literature:
Plant Gene 7 (2016) 42–49 http://dx.doi.org/10.1016/j.plgene.2016.05.001
Molecules 2014, 19, 3450-3459; doi:10.3390/molecules19033450
especially the first of them, from Plant Gene, where the corresponding author is the same as in the reviewed article? The data contained in it in the scope of conducted research and the results obtained significantly overlap with the manuscript presented in this.
In my opinion, the presented work is not novelty, the method is known and used, and already described for the raw materials tested. There are also a lot of stylistic and language errors in the manuscript. The authors also do not use a uniform Latin nomenclature in the names of the studied raw materials (e.g. Fritillariae Cirrhosae Bulbus and Fritillariae cirrhosae bulbus).
Line 168: …have been widely used in the identification of leguminous, Compositae and other traditional Chinese medicines… Was it about: ...have been used to identification of plants of the Fabaceae and Compositae families as well as other plants traditionally used in Chinese medicine
Line 126: … Fritillaria Cirrhosae Bulbus and Fritillaria thunbergii …(also bulbus?).
Reviewer 2 Report
The present paper deals with the development of a combined method for distinguish different Fritillariae bulbus preparations and their adulterants by DNA barcoding and HPLC analysis. The data demonstrate that DNA barcode identification may be useful in the classification of Fritillariae parent sources, and also HPLC analyses provide informations which are useful for differentiation of similar herbal materials. However, the manuscript contains so many mistakes, imprecise phrasing, and many missing information which justify its rejection.
The abstract is not understandable for the readers. It should be clearly describing the different types of preparations (Chuanbeimu, Zhebeimu, Fritillariae bulbus in Chin. Pharmacop.), in the present form it is very troubled.
Although the traditional products of Fritillaria species, products according to the Chinese Pharmacopoeia, and the adequate botanical plant names are clarified in the Introduction part, later the Authors use the inadequate terms, often they mix the names (e.g. “wabuensis”, “Fritillariae walujewil”, “Fritillariae walujewil bulbus”, “Fritillariae Pallidiflorae Bulbus” and “Fritillariae pallidiflorae bulbus”, botanical plant names with or without italic not consequently, etc.). The incorrect product names makes difficult to understand the text. To the historical names of herbal preparations “Chuanbeimu, Zhebeimu” the adequate plant names are not assigned.
HPLC fingerprinting or measurement of the test alkaloids (peimisine, verticine, verticinone, imperialine) was made? The alkaloid content is mentioned in the text, but data not shown, and not discussed in details. Compound identification was made in case of all samples? Table S3 (cited in the MS) is missing!
No voucher specimens of the investigated samples were deposited, which would be very important in such a work.
Abbreviation AD should be explained. Why rhizome bolbostemmae, Coix seed, Cius volvacea were involved in the study, and how they were selected?
Figure 4 and 5 need more explanation. What can be seen?
Incorrect numbering of citation of references can be found several times: “Scientist Herber” not in refs 15-19, and Gui et al. not in ref 30. Further, spaces are needed between text and refs numbers.
The List of References contains a lot of mistakes; it clearly shows the unmindful work.
The title would be better as „Combining DNA Barcoding and HPLC Fingerprints..”
Fritillariae Bulbus or Fritillariae bulbus?
Round 2
Reviewer 1 Report
In future authors should be more careful during preparation of manuscript, both in terms of editorial and scientific matters, while choosing the purpose of research and the legitimacy of its implementation.
Reviewer 2 Report
This improved version can be accepted for publication.